

**Living coccolithophores from the eastern equatorial Indian Ocean during the spring intermonsoon: Indicators of hydrography**

*Jun Sun [1,2,3], Haijiao Liu [1,2,3], Xiaodong Zhang[2,3], Cuixia Zhang[2,3], Shuqun Song[4]

[1] Institute of Marine Science and Technology, Shandong University, 27 Shanda Nan Road, Jinan 250110, PR China

[2] Tianjin Key Laboratory of Marine Resources and Chemistry, Tianjin University of Science and Technology, Tianjin 300457, PR China

[3] College of Marine and Environmental Sciences, Tianjin University of Science and Technology, Tianjin 300457, PR China

[4] CASKey Laboratory of Marine Ecology and Environmental Sciences, Institute of Oceanology, Chinese Academy of Sciences, Qingdao 266071, PR China

*Correspondence to*: Jun Sun (phytoplankton@163.com)

**Abstract.** We studied the biodiversity of autotrophic calcareous coccolithophore assemblages at 30 locations in the eastern equatorial Indian Ocean (EEIO) (80°-94°E, 6°N-5°S) and evaluated the importance of regional hydrology. We found 25 taxa of coccospheres and 17 taxa of coccoliths. The coccolithophore community was dominated by *Gephyrocapsa oceanica*, *Emiliania huxleyi*, *Florisphaera profunda*,*Umbilicosphaera sibogae*, and *Helicosphaera carteri*. The abundance of coccoliths and coccospheres ranged from $0.192\times10^3$ to $161.709\times10^3$ coccoliths $l^{-1}$ and $0.192\times10^3$ to $68.365\times10^3$ cells $l^{-1}$, averaged at $22.658\times10^3$ coccoliths $l^{-1}$ and $9.386\times10^3$ cells $l^{-1}$, respectively. Biogenic PIC, POC, and rain ratio mean values were 0.498 µgC $l^{-1}$, 1.047 µgC $l^{-1}$, and 0.990 respectively. High abundances of both coccoliths and coccospheres in the surface ocean layer occurred north of the equator. Vertically, the great majority of coccoliths and coccospheres were concentrated in water less than 75 m deep. The ratios between the number of coccospheres and free coccoliths across four transects indicated a pattern that varied among different oceanographic settings. The *H'* and *J* values of coccospheres were similar compared with those of coccoliths. Abundant coccolithophores along the equator ) mainly occurred west of 90°E, which was in accordance with the presence of Wyrtki jets (WJs).*F. profunda* was not found in surface water, indicating a stratified and stable water system. *U. irregularis* dominated in the equatorial zone, suggesting oligotrophic water conditions. Coccosphere distribution was explained by environmental variables, indicated by multi-dimensional scaling (MDS) ordination in response variables and principal components analysis (PCA) ordination in explanatory variables. Coccolithophore distribution was related to temperature, salinity, density and chlorophyll *a*.

**1 Introduction**

The Indian Ocean is the world's third largest ocean basin, and it is strongly influenced by the South Asian monsoon system. The warm seawater area in the eastern equatorial Indian Ocean (EEIO) is a large region that influences worldwide climatology and El Niño/Southern Oscillation(ENSO) events (Zhang et al., 2009; Peng et al., 2015). The Indian Ocean dipole is another oceanic phenomenon influencing global oceanographic circulation (Horii et al., 2009). Surface currents in the EEIO are diverse and seasonally dynamic due to monsoon forces. Unlike other ocean basins,the Indian Ocean experiences prevailing semiannual currents (Luyten and Roemmich, 1982; Zhang, 2015). Many currents prevail in the EEIO during the summer and winter monsoon periods. These include the Equatorial undercurrent and the South Java Current (Iskandar, 2009; Peng et al., 2015). There are also currents that exist throughout the year. One example is the Indonesian throughflow (ITF), which is the passageway connecting the Pacific Ocean and Indian Ocean (Ayers et al., 2014). In the spring and fall intermonsoon periods, many surface circulations disappear, and Wyrtki jets (WJs) are the only semi-annual currents present at the equator. The equatorial Indian Ocean is controlled by the eastward WJs (also known as Equatorial Jets)



(Wang, 2015).
Living coccolithophores thrive in the photic water column. Coccolithophores are unicellular microalgal flagellates with
diverse life cycles (Moheimani et al., 2012). They generate external calcified scales (coccoliths) responsible for large areas
of visible "white water" recorded by satellite remote sensing. Coccolithophores are globally distributed and contribute up to
10% of the global phytoplankton biomass (Holligan et al., 1983; Brown and Yoder, 1994; Guptha et al., 2005; Sadeghi et al.,
2012; Hagino and Young, 2015; Oviedo et al., 2015). This calcareous nanoflora usually dominates the open ocean plankton
community (O'Brien et al., 2013; Sun et al., 2014). In its dual functions of biomineralization and photoautotrophy, the
coccolithophore community influences the global carbon cycle and oceanographic parameters (Sun, 2007). Inorganic
calcareous coccoliths can serve as a physical ballast for organic carbon sequestration in the deep ocean (Ziveri et al., 2007;
Bolton et al., 2016; Rembauville et al., 2016). As a consequence, the PIC/POC (particulate inorganic carbon to organic
carbon = "rain ratio"), is a factor explaining biomineralization process impacts on organic production exports.
Coccolithophore assemblages are sensitive to climate variability (Tyrrell, 2008; Silva et al., 2013). Increased $CO_2$
concentrations combined with other factors (e.g., nutrient elements, pH, irradiance, temperature) stimulated cell organic
carbon fixation (photosynthesis) have diminished the rain ratio of coccolithophores (Riebesell et al., 2000; Langer et al.,
2009; Shi et al., 2009; Feng et al., 2008). The coccolithophore cell (coccosphere) is surrounded by several thin layers of
coccoliths, which are useful in reconstructing paleoceanographic history (Guptha et al., 2005; Guerreiro et al.,
2013).Coccolithophore community structure and ecological distributions in the Atlantic Ocean have been documented by
McIntyre et al., (1970), Brown and Yoder, (1994), Baumann et al., (1999), Kinkel et al., (2000), and Shutler et al., (2013).
Pacific Ocean studies have included Okada and Honjo, (1973, 1975), Honjo and Okada, (1974), Okada and McIntyre, (1977),
Houghton and Guptha, (1991), Saavedra-Pellitero, (2011), Saavedra-Pellitero et al., (2014), and López-Fuerte et al., (2015).
Most of the coccolithophore studies were limited to surface waters. Studies on coccolithophores in the Indian Ocean have
been relatively recent compared to Atlantic and Pacific Ocean studies. Coccolithophore studies in the Indian Ocean mainly
include Young (1990), Giraudeau and Bailey (1995), Broerse et al. (2000), Lees (2002), Andruleit (2007), Mohan et al.
(2008), Mergulhao et al. (2013), in regard to nanofossil or living species biogeography in the monsoon season. Relatively
few studies have evaluated the occurrence of living coccolithophores in the water column during the intermonsoon period in
the eastern Indian Ocean. Our three main objectives were to (1) document the abundance, diversity and geographical patterns
of living coccolithophores; (2) explain variations occurring in the nanoflora assemblages; (3) correlate these variations to
regional hydrographic parameters.
**2 Materials and methods**
**2.1 Survey area and sampling strategy**
An initial investigation cruise was conducted in the eastern equatorial Indian Ocean (EEIO) (80°~94°E, 6°N~5°S) (Fig. 1)
onboard R/V "*Shiyan* 1" from March 10th through April 9th, 2012. Seawater samples (400-500 mL) and chlorophyll *a* (Chl*a*)
samples were collected at seven depths from the surface to 200 m using Niskin bottles on a rosette sampler (Sea-Bird
SBE-911 Plus V2). At all the stations, temperature and salinity profile data were determined in situ with the attached sensors
system (conductivity-temperature-depth, CTD).
**2.2 Phytoplankton analysis**
Coccolithophore samples were filtered with a mixed cellulose membrane (25 mm, 0.22 μm) using a Millipore filter system





connected to a vacuum pump under < 20 mm Hg filtration pressure. After room temperature drying in plastic Petri dishes,
the filters were cut and subsequently mounted on glass slides with neutral balsam for microscope examination (Sun et al.,

3    2014).

### 2.3 Size-fractionated Chl*a* analysis

Chl*a* samples were serially filtered using the same filtration system (vacuum < 200 mm Hg) through 20 μm × 20 mm silk net
(micro-class), 2 μm × 20 mm nylon membrane (nano-class) and 0.7 μm × 20 mm Whatman GF/F filters (pico-class). After
filtration, Chl*a* membranes were immediately wrapped in aluminum foil and stored in a freezer -20℃ freezer. In the
laboratory, Chl*a* measurements were made using the fluorescence method of Parsons et al. (1984).

### 2.4 Estimation of coccolith calcite, coccosphere carbon biomass

The cell size biovolume was evaluated from geometric models (Sun and Liu, 2003) and then converted into carbon biomass
(POC, particulate organic carbon) using the formula of Eppley et al. and Guo et al. (Eppley et al., 1970; Guo et al., 2016).
Determinations of calcite-$CaCO_3$ (PIC, particulate inorganic carbon) masses were based on $k_s$ values (shape factor) and
length maximum (diameter, μm) recorded in previous studies (Young and Ziveri, 2000; Yang and Wei, 2003). The PIC/POC
value is a potential rain ratio, which expresses the carbonate flux export to the outside of the euphotic water.As for the
irregularly shaped coccolithophores whose biovolume has rare records, nearly 33% of the species were estimated with
geometric models using SEM pictures from the literature, websites, and this study (Kleijne, 1991; Giraudeau and Bailey,
1995; Cros and Fortuño, 2002; Young et al., 2003). The website can be visited via the access:
http://ina.tmsoc.org/Nannotax3/index.html. It is noted that organic carbon was calculated with the exception of *Gladiolithus*
*flabellatus* and *Reticulofenestra sessilis* by the reason of insufficient records in SEM.

### 2.5 Multivariate analysis

The spatial distribution of coccolithophores and hydrologic data were analyzed using freeware package Ocean Data View
(ODV) 4.7.6 (https://odv.awi.de/en/). Box-whisker plots were prepared by the Golden Software Grapher(LLC, Colorado,
USA  ) 10.3.825. Cluster analysis and non-metric multidimensional scaling (Shen et al., 2010) on coccosphere data (after
square root transformation) were simultaneously implemented using the program package PRIMER 6.0 (Plymouth Routines
In MultivariateEcological Research, developed at the PlymouthMarine Laboratory, United Kingdom). Prior to the above
operations, the raw data were square root transformed. Then, principal component analysis (PCA) considering Euclidean
distance was employed after data transformation and normalization. Significance testing was performed using the Analysis
of Similarities (ANOSIM) analysis. The Similarity Percentages-Species Contributions the Similarity Percentages Routine
(SIMPER) program was used for evaluating the contribution of each species to their sample group. All analyses were
conducted to visualize the relations between phytoplankton abundance data and specific environmental factors.

### 3 Results

### 3.1 Hydrographic features

High temperature and highly saline waters from the west equatorial zone were advected into the east equatorial zone (Fig. 2a,
b). The temperature-salinity (T-S) curve had an inverted-L-shape (Fig. 2c). During the spring monsoon transition period, the
water column was well stratified and quite stable, which is mainly attributed to weak wind-driven surface circulation
compared to the monsoon period (vertical temperature and salinity data not shown). Due to the well stratified water column,



the spring intermonsoon was considered to be the most oligotrophic period (Rixen et al., 1996).
**3.2 Taxonomic composition and characteristics**
Samples of living coccolithophores from the EEIO during the spring intermonsoon period yielded 26 species, representing
25 taxa of coccospheres and 17 taxa of coccoliths. Scanning electron microscope (SEM) photographs of selected species are
shown in Plates I-VI, including several predominant taxa. Among coccolith species, *Gephyrocapsa oceanica*, *Emiliania*
*huxleyi*, *Umbilicosphaera sibogae*, *Helicosphaera carteri*, and *H. hyalina* were most dominant. Coccosphere assemblages
were dominanted by *G. oceanica*, *Florisphaera profunda*, *E. huxleyi*, *Umbellosphaera irregularis*, and *U. sibogae*. *G.*
*oceanica* was overwhelmingly dominant among the coccoliths, with frequency and relative abundance up to 96.5% and
71.76%, respectively. The rest of coccolith species were similar in frequency and abundance. *G. oceanica* and *E. huxleyi* had
high frequencies, with 44.5% and 31%, respectively. *F. profunda* had the highest (up to 40.78%) relative abundance (Fink et
al., 2010).
Coccolith and coccosphere density ranged from $0.192 \times 10^3$ to $161.709 \times 10^3$ coccoliths $l^{-1}$ and $0.192 \times 10^3$ to $68.365 \times 10^3$ cells $l^{-1}$,
averaged at $22.658 \times 10^3$ coccoliths $l^{-1}$ and $9.386 \times 10^3$ coccoliths $l^{-1}$, respectively. The most predominant coccolith species *G.*
*oceanica* was ranged $\sim 154.955 \times 10^3$ coccoliths $l^{-1}$, with a mean value of $16.260 \times 10^3$ coccoliths $l^{-1}$. And the most predominant
coccosphere species was still represented by *G. oceanica*, whose abundance ranged $\sim 24.805 \times 10^3$ cells $l^{-1}$, with average value
$2.458 \times 10^3$ cells $l^{-1}$. The abundances of five dominant coccolith and five coccosphere species are shown in Fig. 3. The other
dominant coccoliths had similar abundances. For the remaining coccosphere species, *G. oceanica* and *U. irregularis* were
more abundant than *E. huxleyi* and *U. sibogae*.
**3.3 Distribution and diversity pattern**
The horizontal distributions of dominant coccoliths and coccospheres are shown in Fig. 4 and Fig. 5. Coccolith abundance
was greatest in three regions: south of Sri Lanka, easternmost Sri Lanka, and southernmost area (Fig. 4). Abundance was
relatively low in the equatorial region. In contrast to the coccoliths, coccospheres were more homogeneous in their
horizontal distributions (Fig. 5).
Dominant coccolithophores abundances along two sections are illustrated in Figs. 6~9. More abundant coccolith species
were restricted to the water column west of 90°E (Fig. 6). Nearly no coccoliths were distributed from the surface down to 50
m along east of 90°E. Dominant coccospheres abundance in section A were mainly represented by *F. profunda* and *U.*
*irregularis* (Fig. 7). These two taxa followed trends similar to the coccoliths. For section B, coccolith abundance was
primarily due to *G. oceanica* (Fig. 8) and abundance was concentrated in the easternmost region. *E. huxleyi* and *U. sibogae*
were mainly distributed in deeper water. *H. hyalina* abundance decreased in deeper and open water and *H. carteri* showed a
plaque pattern. Fig. 9 shows obvious coccosphere abundance in the 75 m water layer of section B, where a deep abundance
maximum was located. *F. profunda* was the dominant coccosphere in the assemblage at section B.
Vertically, numerous dominant coccoliths were confined to the middle layer in the EEIO (Fig. 10). The others reached peak
values at the 50 m water layer, except for *E. huxleyi* and *H. carteri*, whose peak values were located in the 200 m and 100 m
water layers. Coccosphere species increased from the surface towards the middle water, and then decreased towards the
bottom water (Fig. 11). The ratios between coccospheres and free coccoliths were charted along transects (Fig. 12). The ratio
values basically coincided with coccosphere abundance. The ratio reached a maximum in the 40 m layer along sections A
and C. The ratio along section B exhibited a differed trend and its maximum was at the surface layer. The section D ratio was
concurrent with the section C ratio.



**3.4 Estimation of PIC, POC, and rain ratios**

The mean PIC, POC, and rain ratios were 0.498 µgC l⁻¹, 1.047 µgC l⁻¹, and 0.990, respectively. The surface distributions and depth-integrated patterns of PIC, POC, and rain ratio are shown in Fig. 13. We found a dominance of *Oolithotus fragilis* and *G. oceanica* in biogenic PIC. Unlike PIC, POC was mainly contributed by cells of *U. sibogae* and *U. irregularis*. The pattern of PIC and POC appeared to be similar. The surface water of the inner and outer of Sri Lanka section displayed two peaks. In the case of the integral value, PIC and POC were preferentially distributed west of the equator. The depth averaged-rain ratio peak occurred at 80°E-85°E.

In section A, *O. fragilis* contributed about 48% of total PIC, with a maximum value at Station (St.) I405 accounting for 94%. The POC distribution pattern was similar to *U. irregularis* abundance. The maximum rain ratio value occurred east of the equator. In section B, PIC was represented by *F. profunda*. POC and cell abundance showed concurrent trends. Rain ratio had a clear pattern with higher values in the surface and bottom layers.

**3.5 Coccosphere clustering and analysis**

Coccosphere samples at 75 m layer (Deep Chlorophyll Maximum, DCM), where great quantities of coccosphere located, were chosen for the cluster and MDS analysis. The combinations of clustering technique and MDS method are usually conductive to obtain balanced and reliable conclusions in ecological studies (Liu, 2015;Clarke and Warwick, 2001). All samples could be clustered into four groups (Group a, b, c, d). MDS stress values (0.15) lesser than 0.2 give an useful ordination picture, particularly at the lower end of this range (Cox and Cox, 1992;Clarke and Warwick, 2001). ANOSIM analysis revealed remarkable difference (Global R=0.85, p=0.001) among group classification with the exception of Group b-d and Group c-d whose R value < p value (Fink et al., 2010). It is accepted that Global R value larger than 0.5 accounts for significant difference among groups (Liao, 2013). Apparently, localities were basically classified along transects (e.g. Group c included the equatorial localities), whereas some exceptions existed (Fig. 14). Besides, MDS bubble plots for first six dominant coccosphere species were presented in Fig. 14. It is apparently that, Group a and b were mainly composed by dominant coccosphere *G. oceanica*, *F. profunda* and *E. huxleyi*. While Group c was primarily contributed by species *U. sibogae* and *U. irregularis*. Considering Group d only contained two localities, *G. oceanica* dominated the whole group. The SIMPER results were shown in Table 4. It showed the contribution rates of dominant coccosphere in each group.

**4 Discussion**

**4.1 Coccolithophore species diversity and distributionsin the EEIO**

The surface water of eastern Sri Lanka had the greatest coccolith and coccosphere species richness and abundance. The biodiversity indices were much lower around the neighboring waters of Sri Lanka (Fig. 15), suggesting that the local water in that system lacked ecosystemstability. The *H'* and *J* coccospheres values were slightly higher compared with coccolith values (Fig. 16). Therefore, coccosphere aggregations exhibited more diversity than coccoliths. This finding was consistent with that of Guptha et al. (2005). The physical distributions of coccolithophore assemblages in relation to the temperature-salinity are also shown (Figs. 17, 18). The coccoliths represented by *G. oceanica*, *U. sibogae*, *H. carteri* and *H. hyalina* were concentrated in the surface layer characterized by high temperature and low salinity and the bottom euphotic layer characterized by low temperature and high salinity. Conversely, *E. huxleyi* was predominantly distributed in the intermediate layer with moderate temperature and salinity. The coccospheres, *F. profunda* and *E. huxleyi* were mainly found in the deeper euphotic layer where the DCM layer is located. *U. irregularis* and *U. sibogae* had greater abundances in the





surface layer, confirming their preference for oligotrophic conditions.
The POC pattern can be represented by coccosphere abundance. Varied allocation to calcification produced dissimilarities in
the PIC/POC ratios. Large rain ratio values around the Sri Lanka waters predicted a mineral ballast with a drawdown of
biological carbon towards the deep seafloor (Iglesias-Rodriguez et al., 2008; Findlay et al., 2011). We suggest that the rain
ratio (Zondervan et al., 2002) is of great importance in predicting biominerolization and photosynthetic production (Bolton et
al., 2016).
**4.2 Coccolithophore ecological preferences**
Many coccolithophore indicator species were collected in this study allthough several were uncommon. *G. oceanica* is a
representative dominant species that shows preference for eutrophic water (Andruleit et al., 2000). In the surface distribution
of *G. oceanica*, both coccoliths and coccospheres were predominantly distributed in the easternmost waters of Sri Lanka.
This may be due to the nutrients derived from the Andaman Sea. The coccosphere of *U. irregularis* was only common in the
equatorial zone, indicating oligotrophic water conditions there (Kleijne et al., 1989). In the Indian Ocean, eight species of
*Florisphaera* were discovered in deep water (Kahn and Aubry, 2012). We found only one species of *Florisphaera* (*F.*
*profunda*) and it occurred in the disphotic layer below 100 m. As an inhabitant of deep water, *F. profunda* was not found in
surface water layer, indicating a stratified and stable water system. The cosmopolitan taxa, *Calcidiscus leptoporus*, was
detected and its coccoliths peaked at a depth of 200 m at St.I705. *C. leptoporus* is sparsely distributed in the water column,
whereas it predominates in the coccolithophore flora of the sediment owing to its resistance to disintegration (Renaud et al.,
2002).The ratios between the number of coccospheres and free coccoliths across four transects were separately demonstrated
and the vertical distribution patterns were variable. This level of biogeographic variation might be related to regional
hydrographic features. We presumed that coccospheres disintegrated into coccoliths after sinking for a certain distance at
section B. Different circumstances appeared at section A, where a subsurface coccosphere maximum at the 40 m layer
occurred. This finding coincided with the pattern of biological abundance. Ratios in sections C and D were consistent with
ratios observed in the equator section (Monechi et al., 2000).
**4.3 Factors regulating coccolithophore assemblage structure**
Coccolithophore abundance was relatively low during the low wind transition period compared to previous studies
conducted during the monsoon period in the EEIO. The low abundance is due to the gentle winds and low nutrient
availability during the spring intermonsoon season leading to low primary productivity and biomass in the EEIO (Morrison
et al., 1998). The surface coccolithophores were most abundant in the northeastern area where pockets of low-salinity water
plume occur (Fig. 2). This resulted from the inflow of less saline water into the equatorial Indian Ocean from the Bay of the
Bengal and Andaman Seas (Wyrtki, 1961; LaViolette, 1967). The outflows derived from the surface water of the Andaman
Sea become concentrated between the south Nicobar Islands and Sumatra (Rizal et al., 2012). In contrast, a highly saline
water tongue was observed along the equatorial Indian Ocean (west of 90°E), indicating that Wyrtki jets (WJs) prevailed
during the spring intermonsoon period. There was consistency in the nanofloral distribution pattern at the equator (section A,
Figs. 6, 7). The maximum abundance along west of 90°E was probably caused by inflow from WJs considering their ability
to alter the oceanic layer structure. PCA was carried out to examine the relationships among the environmental variables (Fig.
19). Coccolithophore abundance was driven primarily by temperature, salinity, density and Chl*a*. The cluster of
environmental data from sample locations coincided with the grouping of species data (except for a few isolated points). The
most abundant species is shown above each locality symbol. The first three principal components (PC1, PC2, PC3) were





extracted based on eigenvalues larger than 1 and explain 42%, 24%, and 20.2% of the variation, respectively. The cumulative
variances of the three components reached up to 86.2% (PC3 not shown). The eigenvectors of all five principal components
are shown in Table 5. The results of PCA indicated that salinity, density, and pico-Chl*a* had a positive relation with PC1,
whereas a close correlation occurred in Group B that was dominated by *E. huxleyi*and *G. oceanica*. Similarly, temperature,
Chl*a*, micro-Chl*a* and nano-Chl*a* were positively correlated to PC2. Groups C and D, characterized by *U. irregularis*, were
associated with temperature. The majority of localities in Group A (represented by *F. profunda*) were negatively related to
Chl*a* and size-fractionated Chl*a*. Finally, the MDS ordination of coccosphere abundance and the PCA ordination of
environmental variables are in good agreement. This high degree of matching in our study confirmed that the present
explanatory variables (Tezel and Hasırcı, 2013) are appropriate for explaining the biological response variables.
**5 Conclusions**
The coccolithophore assemblage in the EEIO during the spring intermonsoon season was primarily comprised of the
coccoliths *G. oceanica*, *E. huxleyi*, *U. sibogae*, *H. carteri*, and *H. hyalina* and the coccospheres *G. oceanica*, *F. profunda*, *E.*
*huxleyi*, *U. irregularis*, and *U. sibogae*s. The abundance of coccoliths and coccospheres ranged from $0.192 \times 10^3$ ~
$161.709 \times 10^3$ coccoliths $l^{-1}$ and $0.192 \times 10^3$ ~ $68.365 \times 10^3$ cells $l^{-1}$, with an average value of $22.658 \times 10^3$ coccoliths $l^{-1}$ and
$9.386 \times 10^3$ cells $l^{-1}$, respectively. The mean values of biogenic PIC, POC, and the rain ratio were 0.498 µg C $l^{-1}$, 1.047 µg C $l^{-1}$,
and 0.990, respectively. The rain ratio was considered to be of great importance so relative biovolume and carbon biomass
were calculated. Additional studies using direct chemical treatments on coccoliths and coccospheres might establish a
relationship between biovolume conversion and chemical measurements and provide more accurate data.
The localities and coccosphere species were ordered by MDS and all samples were clustered into four groups in the EEIO.
The coccolithophore abundance in this study was relatively low and resulting from the weak winds and minimal nutrient
upwelling compared to previous studies that were conducted during the summer or winter monsoon seasons. During the
spring intermonsoon period, no significant oceanic circulation occurred in the EEIO except for WJs. We inferred that, in the
study area, different coccolithophore species had specific environmental preferences. Thus, coccolithophore species are good
indicators of oceanographic changes. PCA was used to study the correlation among environmental variables, indicating
positive or negative relationships with nanofloral species. Coccosphere distribution was highly correlated to specific
environmental variables. This was shown by the MDS ordination of response variables and PCA ordination of explanatory
variables. Coccolithophores can be used as dynamic indicators of the upper ocean for their sensitivity to environmental
changes. Obtaining knowledge of specific cellular physiological behavior related to global change variables will be a future
challenge. We attempted to evaluate coccolithophore POC contents using a carbon-volume model that was subject to a
degree of error. Future planned studies will involve indoor experiments using axenic cultures of coccolithophores. The cell
POC will be measured using advanced chemical techniques. Carbon evaluation of the field community will then be
compared with direct measurements from controlled laboratory experiments.
*Acknowledgements*. We wish to thank the Dr. Dongxiao Wang and Dr. Yunkai He for providing and processing CTD dataset. Dr. Ying
Wang, Bing Xue and Xiaoqian Li were also appreciated for their constructive comments on the paperwork. This work was supported by
the Natural Science Foundation of China (41276124) andNational Basic Research Program of China (2015CB954002), Science Fund for
University Creative Research Groups in Tianjin (TD12-5003), and the Program for Changjiang Scholars to Jun Sun. It was also partly
supported by the Natural Science Foundation of China (41676112, 41306119, 41306118). The Captain and Crews of R/V *Shiyan1* were



acknowledged for their assistance in sample collection during the cruise, and the Open Cruises from the Natural Science Foundation of
China. We also thank LetPub (www.letpub.com) for its linguistic assistance during the preparation of this manuscript.

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

Table legends:





Table 1 Living coccolithophores composition in the eastern equatorial Indian Ocean during spring intermonsoon period of

2         2012.

Table 2 Predominant species abundance in the eastern equatorial Indian Ocean during spring intermonsoon period of

4         2012.

Table 3 Global test by ANOSIM analysis in coccosphere species matrix.
Table 4 Dominant coccosphere species and their contribution to each group revealed by means of SIMPER analysis.
Table 5 The statistical values by PCA analysis in coccosphere species matrix.




1    Table 1 Living coccolithophores composition in the eastern equatorial Indian Ocean during spring intermonsoon period of

2    2012.

| **Species** | Frequency of occurrence (%fi) | Relative abundance(%P) | Dominance degree(Y) |
|---|---|---|---|
| Dominant coccoliths | | | |
| *Gephyrocapsa oceanica* | 96.5 | 71.76 | 0.6925 |
| *Emiliania huxleyi* | 64.0 | 8.00 | 0.0512 |
| *Umbilicosphaera sibogae* | 62.5 | 6.26 | 0.0391 |
| *Helicosphaera carteri* | 63.5 | 3.50 | 0.0222 |
| *Helicosphaera hyalina* | 61.5 | 3.02 | 0.0186 |
| Dominant coccospheres | | | |
| *Gephyrocapsa oceanica* | 44.5 | 26.18 | 0.2330 |
| *Florisphaera profunda* | 22.0 | 40.78 | 0.1794 |
| *Emiliania huxleyi* | 31.0 | 6.46 | 0.0400 |
| *Umbellosphaera irregularis* | 15.3 | 11.75 | 0.0358 |
| *Umbilicosphaera sibogae* | 30.0 | 4.05 | 0.0243 |




2      Table 2 Predominant species abundance in the eastern equatorial Indian Ocean during spring intermonsoon period of

3                             2012.

| **Dominant coccoliths** | Min, Max (Mean) Units (coccoliths ml$^{-1}$) |
|---|---|
| *Gephyrocapsa oceanica* | -, 154.955 (16.260) |
| *Emiliania huxleyi* | -, 23.706 (1.814) |
| *Umbilicosphaera sibogae* | -, 29.04 (1.418) |
| *Helicosphaera carteri* | -, 7.829 (0.793) |
| *Helicosphaera hyalina* | -, 10.307 (0.685) |
| **Dominant coccospheres** | Min, Max (Mean) Units (cells ml$^{-1}$) |
| *Gephyrocapsa oceanica* | -, 24.805 (2.458) |
| *Florisphaera profunda* | -, 53.845 (3.828) |
| *Emiliania huxleyi* | -, 20.167 (0.606) |
| *Umbellosphaera irregularis* | -, 24.675 (1.103) |
| *Umbilicosphaera sibogae* | -, 3.609 (0.381) |





1             Table 3 Global test by ANOSIM analysis in coccosphere species matrix.

| Pairwise Tests | | | | | |
|---|---|---|---|---|---|
| Groups | R Statistic | Significance level % | Possible permutations | Actual permutations | Number >= observed |
| a, b | 0.687 | 0.1 | 1961256 | 999 | 0 |
| a, c | 0.997 | 0.1 | 38760 | 999 | 0 |
| a, d | 0.999 | 0.8 | 120 | 120 | 1 |
| b, c | 0.862 | 0.2 | 8008 | 999 | 1 |
| b, d | 0.989 | 1.5 | 66 | 66 | 1 |
| c, d | 0.906 | 3.6 | 28 | 28 | 1 |





Table 4 Dominant coccosphere species and their contribution to each group revealed by means of SIMPER analysis.

| Group | Average similarity | Dominant species contribution |
|---|---|---|
| Coccospheres | | |
| d | 40.49 | *Gephyrocapsa oceanica* (99.52) |
| b | 53.78 | *Gephyrocapsa oceanica* (40.38); *Emiliania huxleyi* (28.62); *Oolithotus fragilis* (11.63); *Florisphaera profunda*(7.97); *Helicosphaera carteri*(4.18) |
| c | 59.53 | *Umbellosphaera irregularis*(43.67); *Umbilicosphaera sibogae*(27.06); *Gephyrocapsa oceanica* (10.28); *Helicosphaera hyaline* (8.07); *Emiliania huxleyi* (5.36) |
| a | 61.21 | *Florisphaera profunda*(61.89); *Gephyrocapsa oceanica* (22.20);*Algirosphaera robusta* (7.02) |




1            Table 5 The statistical values by PCA analysis in coccosphere species matrix.

| Eigenvectors | | | | | |
|---|---|---|---|---|---|
| Variable | PC1 | PC2 | PC3 | PC4 | PC5 |
| Temperature | -0.423 | 0.468 | 0.019 | 0.302 | -0.34 |
| Salinity | 0.468 | -0.102 | 0.137 | 0.311 | -0.787 |
| Density | 0.459 | -0.455 | 0.084 | -0.016 | 0.163 |
| Chl*a* | 0.42 | 0.488 | 0.089 | 0.241 | 0.305 |
| Micro | 0.307 | 0.413 | -0.284 | -0.755 | -0.251 |
| Nano | 0.202 | 0.348 | 0.682 | -0.007 | 0.199 |
| Pico | 0.282 | 0.186 | -0.648 | 0.429 | 0.206 |

Legends:
Fig.1.Study area in the eastern equatorial Indian Ocean showing the station locations.
Fig. 2. Sea surface temperature (°C) and salinity in the surveyed area (left); Temperature-salinity (T-S) diagram in the
6       surveyed area, the blue solid line showed an inversed-L-shape of the hydrologic data (right).

Fig. 3.The abundance of dominant coccolithophore species in the eastern equatorial Indian Ocean. (units: coccoliths $l^{-1}$, cells
8       $l^{-1}$)

Fig. 4.The surface distribution of dominant coccoliths (units: coccoliths $l^{-1}$) in the surveyed area.
Fig. 5.The surface distribution of dominant coccospheres (units: cells $l^{-1}$) in the surveyed area.
Fig. 6.Dominant coccolith distributions (units: coccoliths $l^{-1}$) along section A of the surveyed area.
Fig. 7.Dominant coccosphere distributions (units:cells $l^{-1}$) along section A of the surveyed area.
Fig. 8.Dominant coccolith distributions (units: coccoliths $l^{-1}$) along section B of the surveyed area.
Fig. 9.Dominant coccosphere distributions (units: cells $l^{-1}$) along section B of the surveyed area.
Fig. 10.Vertical distributions of dominant coccoliths (units: coccoliths $l^{-1}$)in the surveyed area. (a) Sum; (b) *Gephyrocapsa*
16       *oceanica*; (c) *Emiliania huxleyi*; (d) *Umbilicosphaera sibogae*; (e) *Helicosphaera carteri*; (f) *Helicosphaera hyaline*

Fig. 11.Vertical distributions of dominant coccospheres (units: cells $l^{-1}$)in the surveyed area. (a) Sum; (b) *Gephyrocapsa*
18       *oceanica*; (c) *Florisphaera profunda*; (d) *Emiliania huxleyi*; (e) *Umbellosphaera irregularis*; (f)*Umbilicosphaera*
19       *sibogae*

Fig. 12.The ratio of coccosphere to free coccolith in upper ocean column in the eastern equatorial Indian Ocean. (a): section
21       A; (b): section B; (c): section C; (d): section D

Fig. 13. The horizontal distributions of PIC, POC (units: µgCaCO₃ $l^{-1}$, µgC $l^{-1}$), and rain ratio in the surveyed area. (a)~(c): of
23       surface layer; (d)~(f): of vertically integrated.

Fig. 14 Stations clustered by Bray-Curtis rank similarities and group average linkage (upper);      MDS ordination and its
25       bubble plots for six dominant coccosphere species (below).

Fig. 15.Surface distributions of biodiversity index of coccolithophore in the surveyed area.
Fig. 16.Box and whisker diagrams of biodiversity index of coccolithophore in the surveyed area.
Fig. 17.Scatter plots of coccolith distribution under T-S properties in the surveyed area.
Fig. 18.Scatter plots of coccosphere distribution under T-S properties in the surveyed area.
Fig. 19.Ordination biplot based on PCA analysis among environmental variables of the surveyed area. Notes: group
31       partitions here refered to fig. 13; Chla: chlorophyll *a*, Micro: micro-sized Chla, Nano: nano-sized Chla, Pico: Pico-sized
32       Chla, G.o:*Gephyrocapsa oceanica*, F.p: *Florisphaera profunda*, E.h: *Emiliania huxleyi*, U.i: *Umbellosphaera irregularis*,
33       U.s: *Umbilicosphaera sibogae*, A.r: *Algirosphaera robusta*.

Plate Ⅰ-Ⅵ.





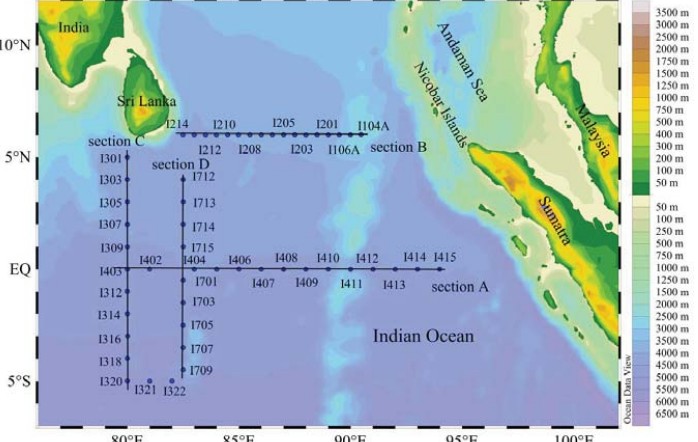

2          Fig. 1. Study area in the eastern equatorial Indian Ocean showing the station locations.





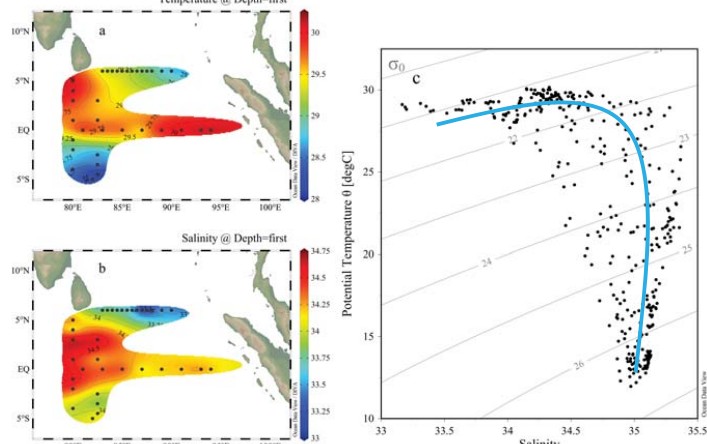

Fig. 2. Sea surface temperature (ºC) and salinity in the surveyed area (left); Temperature-salinity (T-S) diagram in the surveyed area, the blue solid line shows an the inversed-L-shape of the hydrologic data (right).





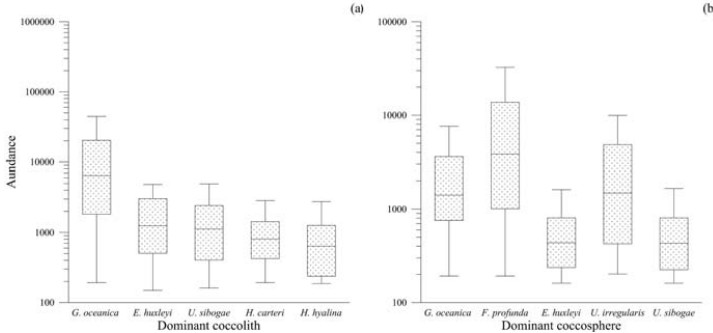

2      Fig. 3. The abundance of dominant coccolithophore species in the eastern equatorial Indian Ocean. (units: coccoliths l$^{-1}$, cells

3                                                                                  l$^{-1}$)




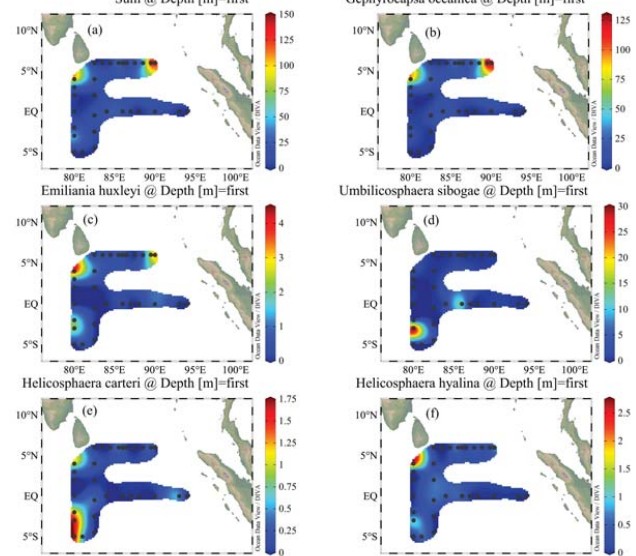

2      Fig. 4. The surface distribution of dominant coccoliths (units: coccoliths l⁻¹) in the surveyed area.




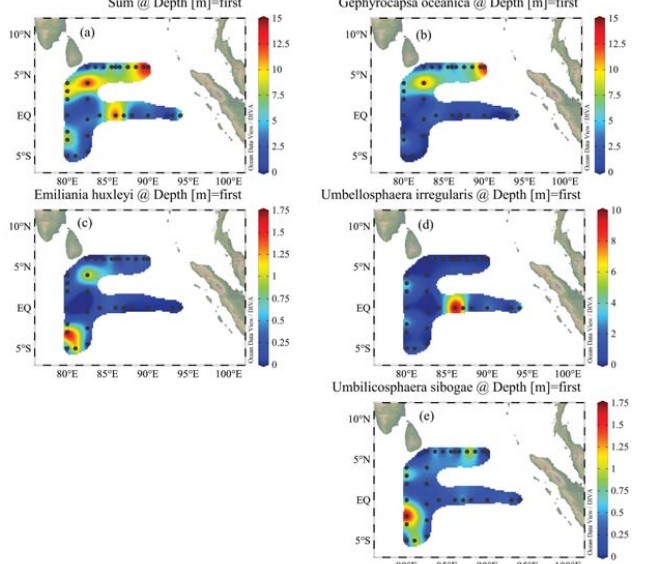

2          Fig. 5. The surface distribution of dominant coccospheres (units: cells l⁻¹) in the surveyed area.



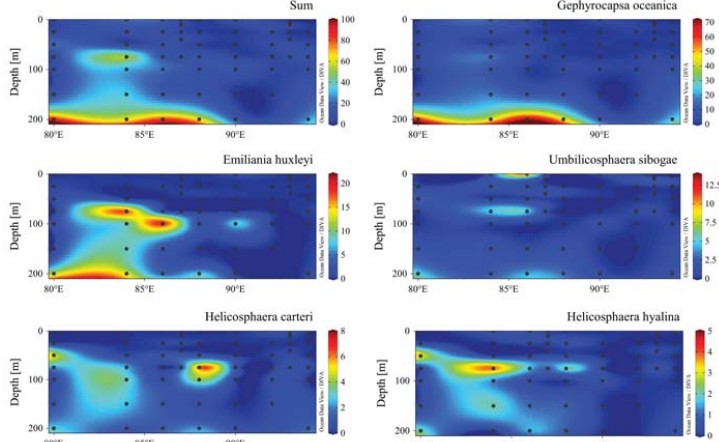

2    Fig. 6. Dominant coccolith distributions (units: coccoliths l⁻¹) along section A of the surveyed area.





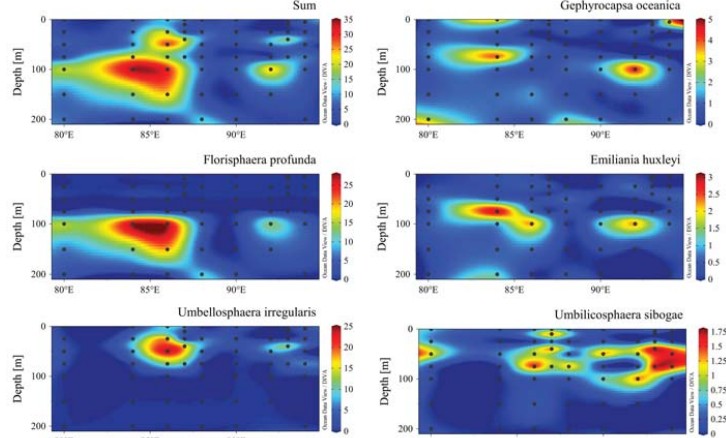

Fig. 7. Dominant coccosphere distributions (units:cells l⁻¹) along section A of the surveyed area.





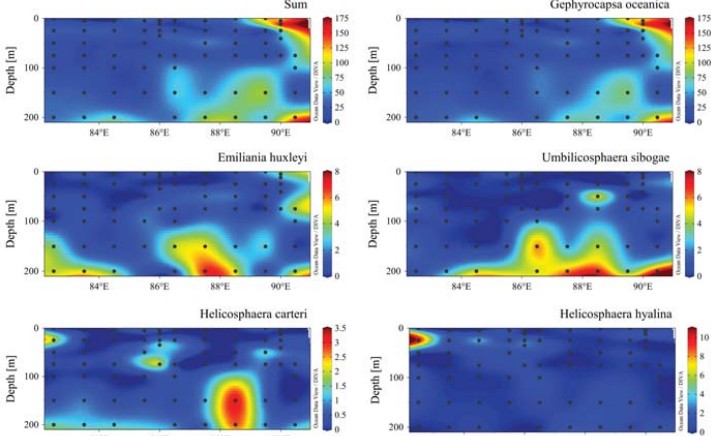

2      Fig. 8. Dominant coccolith distributions (units: coccoliths l$^{-1}$) along section B of the surveyed area.



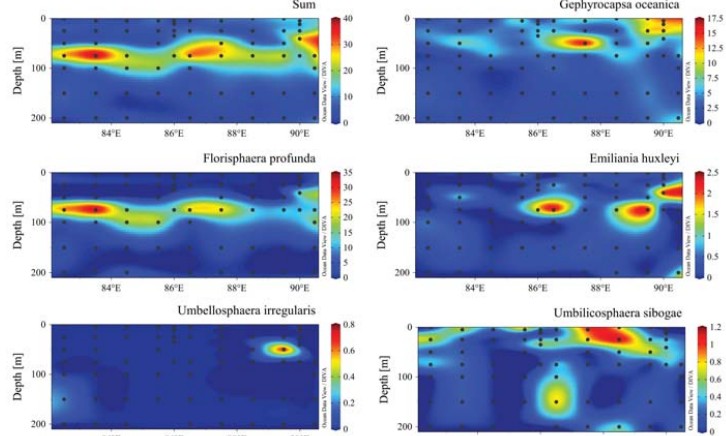

2          Fig. 9. Dominant coccosphere distributions (units: cells l⁻¹) along section B of the surveyed area.



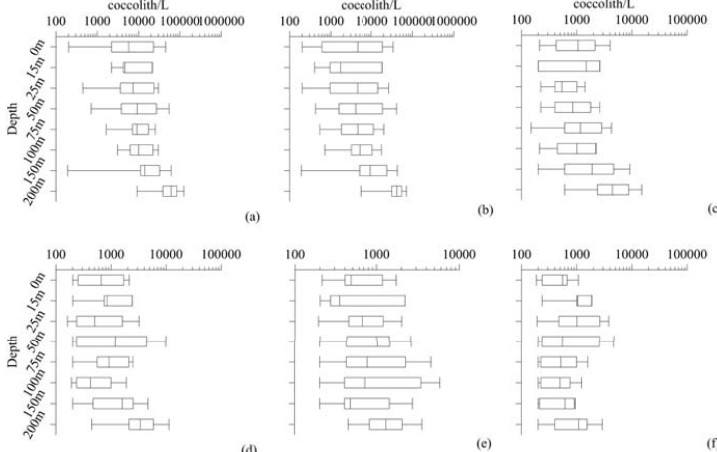

2    Fig. 10. Vertical distributions of dominant coccoliths (units: coccoliths l⁻¹)in the surveyed area. (a) Sum; (b) *Gephyrocapsa*

3       *oceanica*; (c) *Emiliania huxleyi*; (d) *Umbilicosphaera sibogae*; (e) *Helicosphaera carteri*; (f) *Helicosphaera hyaline*




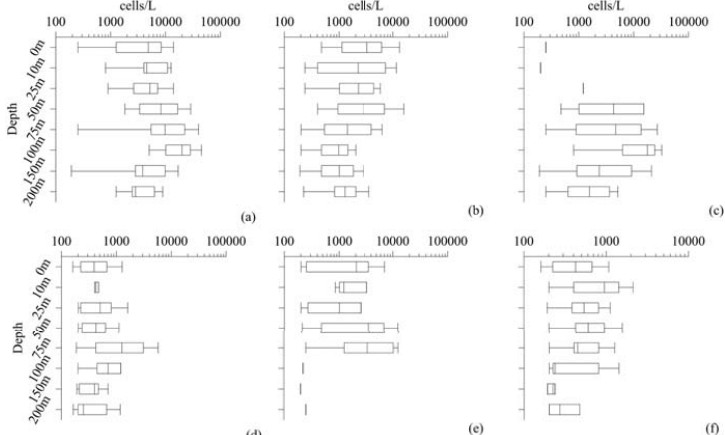

2     Fig. 11. Vertical distributions of dominant coccospheres (units: cells $l^{-1}$)in the surveyed area. (a) Sum; (b) *Gephyrocapsa*

3     *oceanica*; (c) *Florisphaera profunda*; (d) *Emiliania huxleyi*; (e) *Umbellosphaera irregularis*; (f)*Umbilicosphaera sibogae*



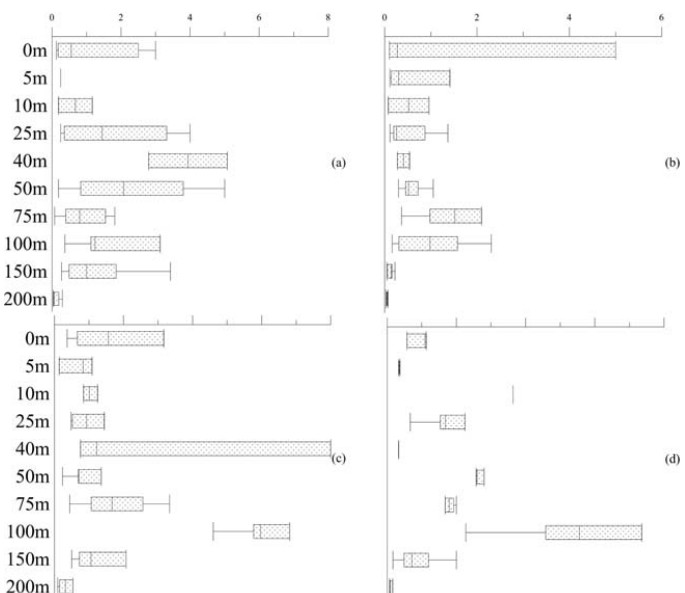

2  Fig. 12. The ratio of coccosphere to free coccolith in upper ocean column in the surveyed area. (a): section A; (b): section B; (c):
3                                        section C; (d): section D



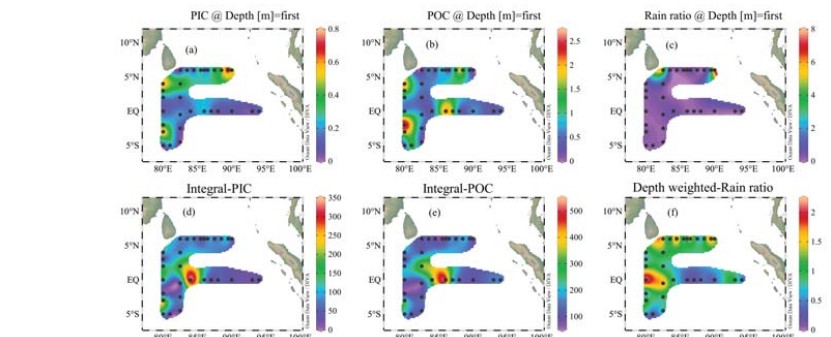

2    Fig. 13. The horizontal distributions of PIC, POC (units:μgC l⁻¹), and rain ratio in the surveyed area. (a)~(c): of surface layer;

3                                            (d)~(f): of vertically integrated.





6      Fig. 14. Stations clustered by Bray-Curtis rank similarities and group average linkage (upper); MDS ordination and its

7          bubble plots for six dominant coccosphere species (below).





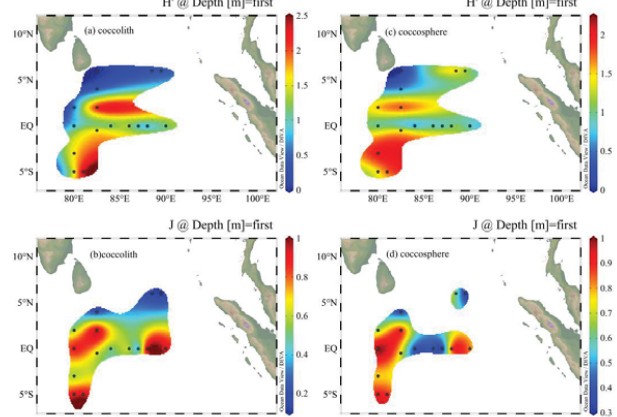

2  Fig. 15. Surface distributions of biodiversity index of coccolithophore in the surveyed area.





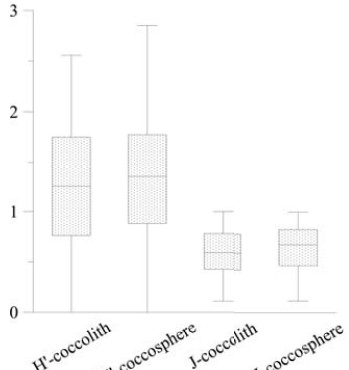

2        Fig. 16. Box and whisker diagrams of biodiversity index of coccolithophore in the surveyed area.





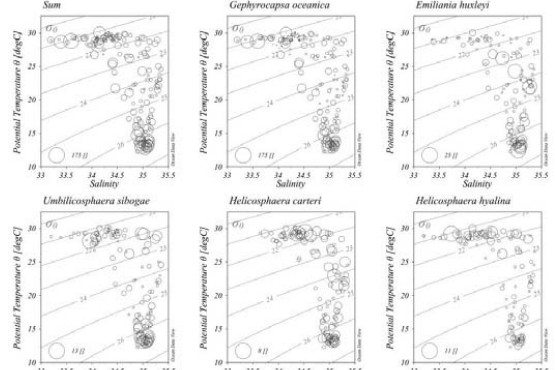

Fig. 17. Scatter plots of coccolith distribution under T-S properties in the surveyed area.



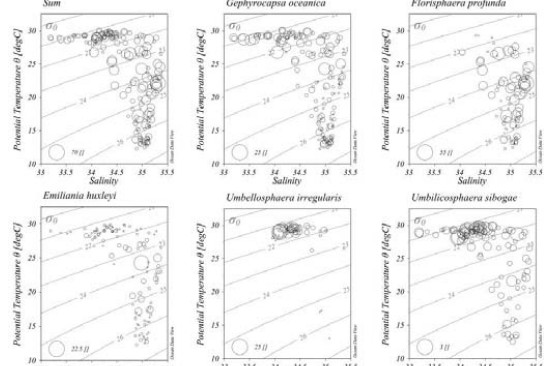

2    Fig. 18. Scatter plots of coccosphere distribution under T-S properties in the surveyed area.





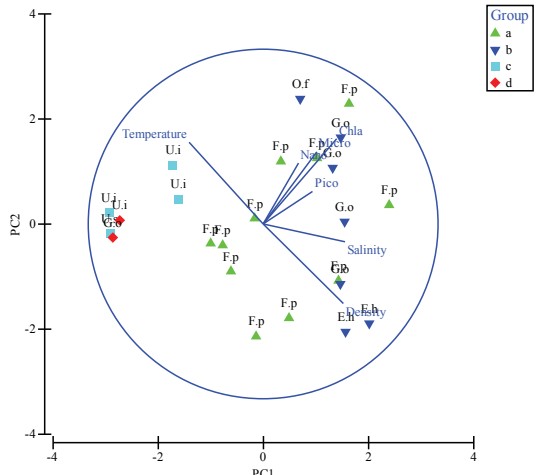

2    Fig.19. Ordination biplot based on PCA analysis among environmental variables of the surveyed area. Note: group partitions

3        here refer to fig. 13; Chla: chlorophyll, Micro: micro-sized Chla, Nano: nano-sized Chla, Pico: Pico-sized Chla,

4    G.o:*Gephyrocapsa oceanica*, F.p: *Florisphaera profunda*, E.h: *Emiliania huxleyi*, U.i: *Umbellosphaera irregularis*, U.s:

5                *Umbilicosphaera sibogae*, A.r: *Algirosphaera robusta*.



Plate Ⅰ. Noëlaerhabdaceae: *Emiliania* & *Gephyrocapsa*

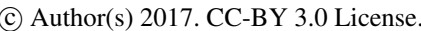

*E. huxleyi* type A overcalcified *G. oceanica*
*G. oceanica* coccolith
*G. oceanica* collapsed



1          Plate Ⅱ . Umbellosphaeraceae: *Umbellosphaera*

3    *U. irregularis*

5    *U. irregularis*

7    *U. tenuisU. tenuis* type I



1                 Plate Ⅲ. Calcidiscaceae: *Umbilicosphaera* &*Calcidiscus*

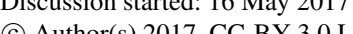

*U.hulburtiana*



*U. foliosaU. sibogaeU.*sp. 1
*U.*sp. 1 cell collapsed
*U.*sp. 1 coccolith detached
*U.*sp. 2               *C. leptoporus*



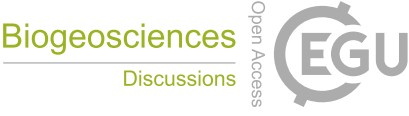

Plate IV.*Reticulofenestra&Ceratolithus&Pontosphaera&Discosphaera*

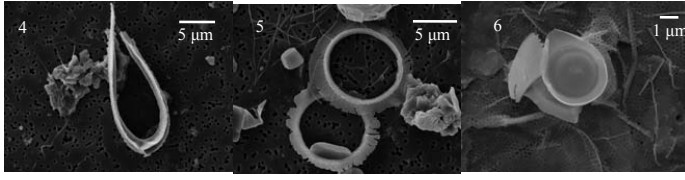

*Reticulofenestra* sp. 1        *Reticulofenestra* sp. 2

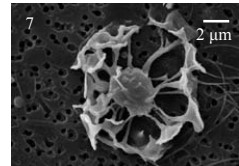

*C. cristatus* CER *telesmus* type*C. cristatus* HET coccolithomorpha type*P. syracusana*
*D. tubifera*



1             Plate Ⅴ. Syracosphaeraceae: *Syracosphaera*

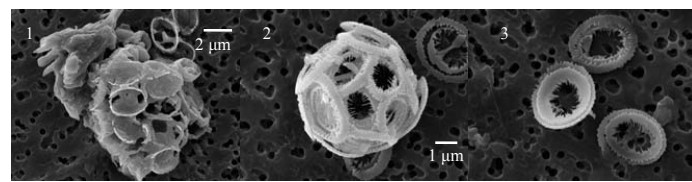

3     Cell disintegrated *S. histrica*





1              Plate Ⅵ. Mixed group

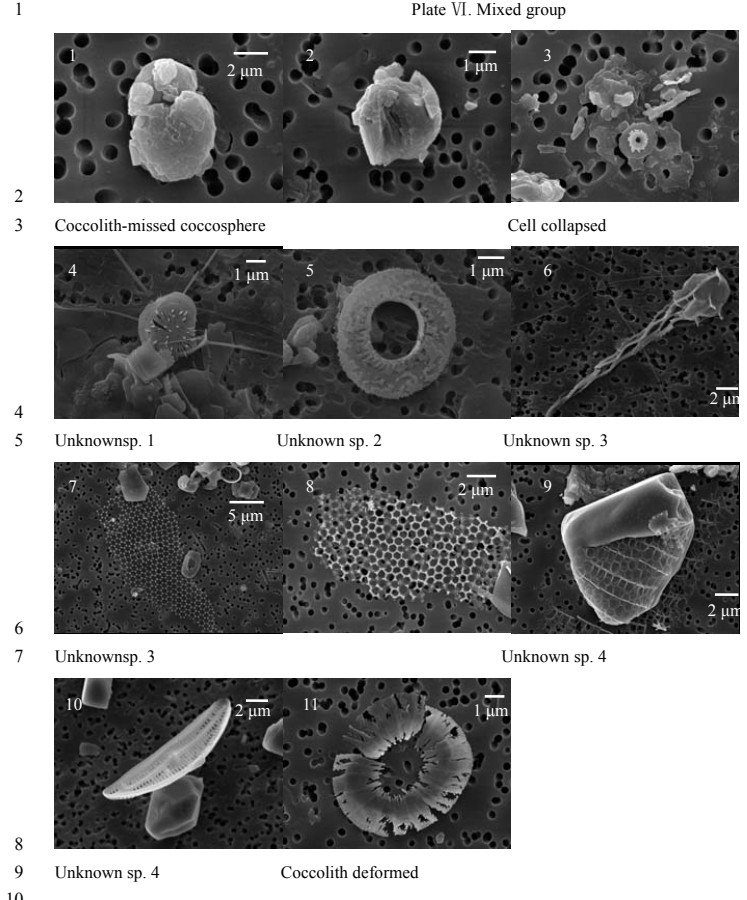

Coccolith-missed coccosphere                    Cell collapsed
Unknownsp. 1            Unknown sp. 2          Unknown sp. 3
Unknownsp. 3                            Unknown sp. 4
Unknown sp. 4          Coccolith deformed

