# Peer review of "Living coccolithophores from the eastern equatorial Indian Ocean"

_Biogeosciences, 2017_

## Referee Comment (RC1) · Anonymous Referee #1 · 26 Jun 2017

Reviewer Comments: The authors have attempted to use of living coccolithophores as indicators of hydrography. Authors have tried to relate/use statistical methods to validate their hypothesis. However, there are many weaknesses in the manuscript. I will comment one by one which must be addressed before publication can be considered. Major points Abstract: The abstract should be more precise. It should address the major outcomes of the paper. The current abstract is too simple and it is difficult for the reader to understand what authors are keen to convey. Introduction: The introduction is very weakly written. First few lines in the Introduction address global issue which is irrelevant for the current subject of the manuscript. The first paragraph can be written after the introduction or in the methods under the heading hydrographic settings. Introduction should start with Second paragraph beginning with introduction of coccolithophores. Introduction to coccolithophores should be more precise. For ex. "Coccolithophores are unicellular microalgal flagellates with diverse life cycle"... authors should explain what kind of life cycle they have. The white water mentioned in the line no 12, happens during bloom condition, calcareous nannoflora usually dominate in the open ocean plankton community............. There are many authors who have described where coccolithophores usually dominate. Line 23- coccolithophore cell is not a coccosphere. Please see definitions of these terms in the Young et al., 1997- 'Guidelines for coccolith and calcareous nannofossil terminology'. Also, as written in line 23, coccoliths on spheres are not used for paleoceanography, it should be coccoliths preserved in the sediments and authors should be describe how they are utilized Line 25- Community structure and ecological distributions in the Atlantic Ocean have been documented by McIntyre et al., .......... etc. (1) McIntyre et al studied nannoplankton in the Pacific and not Atlantic. (2) Reference of Baumann et al., (1999) is not in the reference list. Similarly, Honjo and Okada (1974) reference is missing in the reference. Line 29- Most of the coccolithophore studies were limited to surface waters is not true. There are many recent studies carried out which are not listed in the introduction. Author listed references are both from sediment and water. Authors should refer recent published papers and write introduction more precisely including recent references. Line 30-33, all studies listed here are not carried out during monsoon. For ex. Mohan et al studied ecology of coccolithophores in the Indian Sector of the Southern Ocean during austral summer.... Not during the monsoon season. Page 3-Line-1-4. Objectives should be more precise.

Materials and Methods The methodology needs to be explicit. It is difficult to comprehend how authors did all the analysis (chlorophyll a, phytoplankton, PIC, POC) in 400-500 ml water. Line 12-15- I wonder, is it a phytoplankton analysis or coccolithophores? How much water was filtered? It sounds like samples were analyzed on light microscope, if so authors should give light microscope images along with SEM images. How many coccospheres and coccoliths counted at each station? How coccolithophore

abundance was calculated? How much water was filtered for size fractionated chl-a, why size fractionated chla analysis was carried out? How PIC and POC was measured? Nothing is written about PIC and POC measurements. I am not sure if the method described here can give good estimates of coccolith calcite or coccosphere carbon biomass? Authors should use statistical tools which are relevant to the study.

Results and discussion Results are very weakly written. Hydrographic features should be more informative. Authors provided 19 figures and 4 tables but this data is not discussed properly. I am not sure if all this statistical analysis is essential to talk about ecological preferences. And, the Coccolithophore ecological preferences which are given in the manuscript are not new. The discussion is somewhat misleading. Authors tried to provide information on factors affecting coccolithophore assemblage structure without providing vertical temperature, salinity, nutrient and other necessary data. Authors have reported only few species. Probably, water filtered was not enough to study coccolithophores in water samples (300-400ml or less than that?-as written in methods, this is grossly less or they need to check their records). For providing ecological preferences, authors should use all the physico-chemical parameters (surface and vertical depth) and draw firm conclusion. References Some references are missing the reference list which is listed in the introduction and discussion. Authors should check all references again and cross check with the text. This manuscript requires gross revision and much more additional information to improve it further. Manuscript should focus on key points and should have strong hypothesis. I have pointed out some, corrections in the text which needs to be answered. The quality of language used is below standard and I don't think this manuscript will be of publication quality of BGD even after revision.

---

## Author Comment (AC1) · 1 Jul 2017

Dear editor: Thank you for the comments on our manuscript. According to the first reviewer's comments, we have made the revision as follows: 1. The abstract has been revised according to the comment. 2. The introduction section has been revised by adding recently published paper more precisely. 3. The material and methods has already been supplemented. Most notably, the currently estimated POC and PIC represented for coccolithophore organic carbon and inorganic carbon specifically. 4. The results and discussion are rewritten. T 5. The distribution of coccolithophores had been concluded into the conclusion section. The revised manuscript has been

uploaded to BGD homepage.

Thanks for your sending back those comments and also take our regarding to the other reviewers. I am waiting for your next step for processing our manuscript.

Your Sincerely Jun Sun 7/1/2017

Please also note the supplement to this comment: https://www.biogeosciences-discuss.net/bg-2017-112/bg-2017-112-AC1-supplement.pdf

**Supplement:**

**Living coccolithophores from the eastern equatorial Indian Ocean during the spring intermonsoon: Indicators of hydrography**

*Jun Sun [1, 2,3], Haijiao Liu [1,2,3], Xiaodong Zhang [2,3], Cuixia Zhang [2,3], Shuqun Song [4]

[1] Institute of Marine Science and Technology, Shandong University, 27 Shanda Nan Road, Jinan 250110, PR China

[2] Tianjin Key Laboratory of Marine Resources and Chemistry, Tianjin University of Science and Technology, Tianjin 300457, PR China

[3] College of Marine and Environmental Sciences, Tianjin University of Science and Technology, Tianjin 300457, PR China

[4] CAS Key Laboratory of Marine Ecology and Environmental Sciences, Institute of Oceanology, Chinese Academy of Sciences, Qingdao 266071, PR China

*Correspondence to*: Jun Sun (phytoplankton@163.com)

**Abstract.** We studied the biodiversity of autotrophic calcareous coccolithophore assemblages at 30 locations in the eastern equatorial Indian Ocean (EEIO) (80°-94°E, 6°N-5°S) and evaluated the importance of regional hydrology. We found 25 taxa of coccospheres and 17 taxa of coccoliths. The coccolithophore community was dominated by *Gephyrocapsa oceanica*, *Emiliania huxleyi*, *Florisphaera profunda*, *Umbilicosphaera sibogae*, and *Helicosphaera carteri*. The abundance of coccoliths and coccospheres ranged from $0.192 \times 10^3$ to $161.709 \times 10^3$ coccoliths $l^{-1}$ and $0.192 \times 10^3$ to $68.365 \times 10^3$ cells $l^{-1}$, averaged at $22.658 \times 10^3$ coccoliths $l^{-1}$ and $9.386 \times 10^3$ cells $l^{-1}$, respectively. Biogenic PIC, POC, and rain ratio mean values were 0.498 µgC $l^{-1}$, 1.047 µgC $l^{-1}$, and 0.990 respectively. High abundances of both coccoliths and coccospheres in the surface ocean layer occurred north of the equator. Vertically, the great majority of coccoliths and coccospheres were concentrated in water less than 75 m deep. The ratios between the number of coccospheres and free coccoliths across four transects indicated a pattern that varied among different oceanographic settings. The *H'* and *J* values of coccospheres were similar compared with those of coccoliths. Abundant coccolithophores along the equator mainly occurred west of 90°E, which was in accordance with the presence of Wyrtki jets (WJs).

zone, suggesting oligotrophic water conditions. Coccosphere distribution was explained by environmental variables, indicated by multi-dimensional scaling (MDS) ordination in response variables and principal components analysis (PCA) ordination in explanatory variables. Coccolithophore distribution was related to temperature, salinity, density and chlorophyll *a*.

**1 Introduction**

The Indian Ocean is the world's third largest ocean basin, and it is strongly influenced by the South Asian monsoon system. The warm seawater area in the eastern equatorial Indian Ocean (EEIO) is a large region that influences worldwide climatology and El Niño/Southern Oscillation (ENSO) events (Zhang et al., 2009; Peng et al., 2015). The Indian Ocean dipole is another oceanic phenomenon influencing global oceanographic circulation (Horii et al., 2009). Surface currents in the EEIO are diverse and seasonally dynamic due to monsoon forces. Unlike other ocean basins, the Indian Ocean experiences prevailing semiannual currents (Luyten and Roemmich, 1982; Zhang, 2015). Many currents prevail in the EEIO during the summer and winter monsoon periods. These include the Equatorial undercurrent and the South Java Current (Iskandar, 2009; Peng et al., 2015). There are also currents that exist throughout the year. One example is the Indonesian throughflow (ITF), which is the passageway connecting the Pacific Ocean and Indian Ocean (Ayers et al., 2014). In the spring and fall intermonsoon periods, many surface circulations disappear, and Wyrtki jets (WJs) are the only semi-annual currents present at the equator. The equatorial Indian Ocean is controlled by the eastward WJs (also known as Equatorial Jets) (Wang, 2015).

Living coccolithophores thrive in the photic water column. Coccolithophores are unicellular microalgal flagellates with diverse life cycles (alternating diploid - haploid stage) belonging to marine nanoplankton (Moheimani et al., 2012; Taylor et al., 2016). Life phase transitions can easily occur in natural assemblages when nutrient level changed (Taylor et al., 2017). They generate external calcified scales (coccoliths) responsible for large areas of visible "white water" recorded by satellite remote sensing. The coccolithophore cell is surrounded by several thin layers of coccoliths. Coccolithophores are globally distributed and contribute up to 10% of the global phytoplankton biomass (Holligan et al., 1983; Brown and Yoder, 1994; Guptha et al., 2005; Sadeghi et al., 2012; Hagino and Young, 2015; Oviedo et al., 2015). This calcareous nanoflora usually dominates the open ocean plankton community

(O'Brien et al., 2013; Sun et al., 2014). In its dual functions of biomineralization and photoautotrophy, the coccolithophore community influences the global carbon cycle, sulphur cycle and oceanographic parameters (Sun, 2007; Taylor et al., 2017). Inorganic calcareous coccoliths can serve as a physical ballast for organic carbon sequestration in the deep ocean (Ziveri et al., 2007; Bolton et al., 2016; Rembauville et al., 2016). As a consequence, the PIC/POC (particulate inorganic carbon to organic carbon = "rain ratio"), is a factor explaining biomineralization process impacts on organic production exports. Coccolithophore assemblages are sensitive to climate variability (Tyrrell, 2008; Silva et al., 2013). Increased $CO_2$ concentrations combined with other factors (e.g., nutrient elements, pH, irradiance, temperature) stimulated cell organic carbon fixation (photosynthesis) have diminished the rain ratio of coccolithophores (Feng et al., 2008; Langer et al., 2009; Riebesell et al., 2000; Shi et al., 2009;). These calcifying nanoplankton is negatively affected by ocean acidification with decreased carbonate availability especially in colder water realm (Oviedo et al., 2017; Smith et al., 2017). Herein, the response of coccolithophore ecophysiology to environmental change has aroused a big concern (Poulton et al., 2017). The coccolithophore cell (coccosphere) is surrounded by several thin layers of coccoliths, When detached coccoliths were exported to the deep sediment, whichthey provided an ideal tool to are useful in reconstructing paleoenvironmental changepaleoceanographic history, e.g. sea-surface temperature, mixed layer and nutricline (Ferreira et al., 2017; Guerreiro et al., 2013; Guptha et al., 2005; Laprida et al., 2017). Coccolithophore geographical distributions interact with environment conditions, thus making them useful in paleoenvironmental reconstructions (Laprida et al., 2017). Coccolithophore community structure and ecological distributions in the Atlantic Ocean have been documented by McIntyre et al., (1970), Brown and Yoder, (1994), Baumann et al., (1999), Kinkel et al., (2000), and Shutler et al., (2013). Pacific Ocean studies have included McIntyre et al., (1970), Okada and Honjo, (1973, 1975), Honjo and Okada, (1974), Okada and McIntyre, (1977), Houghton and Guptha, (1991), Saavedra-Pellitero, (2011), Saavedra-Pellitero et al., (2014), and López-Fuerte et al., (2015). Most of the coccolithophore studies were limited to surface waters.

The Indian Ocean is the world's third largest ocean basin, and it is strongly influenced by the South Asian monsoon system. The warm seawater area in the eastern equatorial Indian Ocean (EEIO) is a large region that influences worldwide climatology and El Niño/Southern Oscillation (ENSO) events (Zhang et al., 2009; Peng et al., 2015). The Indian Ocean dipole is another oceanic phenomenon influencing global oceanographic circulation (Horii et al., 2009). Surface currents in the EEIO are

diverse and seasonally dynamic due to monsoon forces. Unlike other ocean basins, the Indian Ocean experiences prevailing semiannual currents (Luyten and Roemmich, 1982; Zhang, 2015). Many currents prevail in the EEIO during the summer and winter monsoon periods. These include the Equatorial undercurrent and the South Java Current (Iskandar, 2009; Peng et al., 2015). There are also currents that exist throughout the year. One example is the Indonesian throughflow (ITF), which is the passageway connecting the Pacific Ocean and Indian Ocean (Ayers et al., 2014). In the spring and fall intermonsoon periods, many surface circulations disappear, and Wyrtki jets (WJs) are the only semi-annual currents present at the equator. The equatorial Indian Ocean is controlled by the eastward WJs (also known as Equatorial Jets) (Wang, 2015).

Studies on coccolithophores in the Indian Ocean have been relatively recent compared to Atlantic and Pacific Ocean studies. Coccolithophore studies in the Indian Ocean mainly include Young (1990), Giraudeau and Bailey (1995), Broerse et al. (2000), Lees (2002), Andruleit (2007), Mohan et al. (2008), Mergulhao et al. (2013), in regard to nanofossil or living species biogeography in the monsoon season. Relatively few studies have evaluated the occurrence of living coccolithophores in the water column during the intermonsoon period in the eastern Indian Ocean. Our three main objectives were to (1) document the abundance, diversity and geographical patterns of living coccolithophores; (2) explain variations occurring in the nanoflora assemblages; (3) correlate these variations to regional hydrographic parameters.

**2 Materials and methods**

**2.1 Survey area and sampling strategy**

An initial investigation cruise was conducted in the eastern equatorial Indian Ocean (EEIO) (80°~94°E, 6°N~5°S) (Fig. 1) onboard R/V "*Shiyan* 1" from March 10[th] through April 9[th], 2012. Seawater was collected at seven depths from the surface to 200 m using Niskin bottles on a rosette sampler (Sea-Bird SBE-911 Plus V2). At all the stations, temperature and salinity profile data were determined in situ with the attached sensors system (conductivity-temperature-depth, CTD).

**2.2  Coccolithophore analysis**

Coccolithophore samples were filtered 400-500 ml with a mixed cellulose membrane (25 mm, 0.22 μm)

using a Millipore filter system connected to a vacuum pump under < 20 mm Hg filtration pressure. After room temperature drying in plastic Petri dishes, the filters were cut and subsequently mounted on glass slides with neutral balsam for polarized microscope (Motic, BA300POL.) examination (Sun et al., 2014). Totally at least 400 fields was counted by the standard of 30 coccospheres and 50 coccoliths enumerated. The coccolithophore abundance was finally calculated following the formula in Sun et al. (2014).

**2.3 Size-fractionated Chl*a* analysis**

[revised manuscript text omitted]

temperature and salinity. The coccospheres, *F. profunda* and *E. huxleyi* were mainly found in the deeper euphotic layer where the DCM layer is located. *U. irregularis* and *U. sibogae* had greater abundances in the surface layer, confirming their preference for oligotrophic conditions. There was a peculiar oceanographic phenomenon at St. I316 characterized by surface lowest temperature and highest salinity, where the coccoliths of *U. sibogae* and *H. carteri* were predominantly occupied (Fig. 4). *F. profunda* was only distributed below 50 m at St. I316, indicating a stratified and stable water locally. In other words, this peculiar hydrology was not caused by vertical upwelling, maybe water mass advection instead. It is very hard to identify what kinds of currents created this peculiar biophysical distribution, after all water currents are not prosperous during intermonsoon period.

The POC pattern can be represented by coccosphere abundance. Varied allocation to calcification produced dissimilarities in the PIC/POC ratios. Large rain ratio values around the Sri Lanka waters predicted a mineral ballast with a drawdown of biological carbon towards the deep seafloor (Iglesias-Rodriguez et al., 2008; Findlay et al., 2011). We suggest that the rain ratio (Zondervan et al., 2002) is of great importance in predicting biominerolization and photosynthetic production (Bolton et al., 2016).

**4.2 Coccolithophore ecological preferences**

Many coccolithophore indicator species were collected in this study although several were uncommon. *G. oceanica* is a representative dominant species that shows preference for eutrophic water (Andruleit et al., 2000). In the surface distribution of *G. oceanica*, both coccoliths and coccospheres were predominantly distributed in the easternmost waters of Sri Lanka. This may be due to the nutrients derived from the Andaman Sea. The coccosphere of *U. irregularis* was only common in the equatorial zone, indicating oligotrophic water conditions there (Kleijne et al., 1989). In the Indian Ocean, eight species of *Florisphaera* were discovered in deep water (Kahn and Aubry, 2012). We found only one species of *Florisphaera* (*F. profunda*) and it typically occurred in the disphotic layer below 100 m. As an inhabitant of deep water, *F. profunda* hardly occurred inwas not found in surface water layer unless with the appearance of vertical upwelling, indicating a stratified and stable water system. The coccoliths of *U. sibogae* and *H. carteri* maxima were found at St. I316 indicating that these species showed affinities to low temperature and high salinity water. 
[revised manuscript text omitted]

Baumann, K. H., Cepek, M., and Kinkel, H.: Coccolithophores as indicators of ocean water masses, surface-water temperature, and paleoproductivity—examples from the South Atlantic, in: Use of Proxies in Paleoceanography, 117-144, Springer, doi10.1007/978-3-642-58646-0_4, 1999.

Bolton, C. T., Hernández-Sánchez, M. T., Fuertes, M.-Á., González-Lemos, S., Abrevaya, L., Mendez-Vicente, A., Flores, J.-A., Probert, I., Giosan, L., and Johnson, J.: Decrease in coccolithophore calcification and $CO_2$ since the middle Miocene, Nat. Commun., 7, doi: 10.1038/ncomms10284, 2016.

Broerse, A., Brummer, G.-J., and Van Hinte, J.: Coccolithophore export production in response to monsoonal upwelling off Somalia (northwestern Indian Ocean), Deep-Sea Res. Pt. II: Topical Studies in Oceanography, 47, 2179-2205, 2000.

Brown, C., and Yoder, J.: Distribution pattern of coccolithophorid blooms in the western North Atlantic Ocean, Cont. Shelf Res., 14, 175-197, 1994.

Clarke, K. R., and Warwick, R.M.: Change in marine communities: an approach to statistical analysis and interpretation, Plymouth, UK: Primer     E, 2001.

Cox, M. A., and Cox, T.: Interpretation of Stress in non-metric multidimensional scaling, Statistica Applicata, 4, 611-618, 1992.

Cros, L., and Fortuño, J. M.: Atlas of northwestern Mediterranean coccolithophores, Sci. Mar., 66, 1-182, 2002.

Eppley, R. W., Reid, F., and Strickland, J.: Estimates of phytoplankton crop size, growth rate, and primary production, Calif. Univ. Scripps Inst. Oceanogr. Bull., 1970.

Feng, Y., Warner, M. E., Zhang, Y., Sun, J., Fu, F.-X., Rose, J. M., and Hutchins, D. A.: Interactive effects of increased pCO2, temperature and irradiance on the marine coccolithophore *Emiliania huxleyi* (Prymnesiophyceae), Eur. J. Phycol., 43, 87-98, doi: 10.1080/09670260701664674, 2008.

Ferreira, J., Mattioli, E., and Van de Schootbrugge, B.: Palaeoenvironmental vs. evolutionary control on size variation of coccoliths across the Lower-Middle Jurassic, Palaeogeogr. Palaeocl., 465, 177-192, doi: 10.1016/j.palaeo.2016.10.029, 2017.

Findlay, H. S., Calosi, P., and Crawfurd, K.: Determinants of the PIC: POC response in the coccolithophore *Emiliania huxleyi* under future ocean acidification scenarios, Limnol. and Oceanogr., 56, 1168-1178, doi:10.4319/lo.2011.56.3.1168, 2011.

[revised manuscript text omitted]

Poulton, A. J., Holligan, P. M., Charalampopoulou, A., and Adey, T. R.: Coccolithophore ecology in

the tropical and subtropical Atlantic Ocean: New perspectives from the Atlantic meridional transect (AMT) programme, Prog. Oceanogr., doi.org/10.1016/j.pocean.2017.01.003, 2017.

[revised manuscript text omitted]

*E. huxleyi* type A overcalcified*G. oceanica*

[Figure]

*G. oceanica* coccolith

[Figure]

*G. oceanica* collapsed

Plate Ⅱ. Umbellosphaeraceae: *Umbellosphaera*

[Figure]

*U. irregularis*

[Figure]

*U. irregularis*

[Figure]

*U. tenuisU. tenuis* type I

Plate Ⅲ. Calcidiscaceae: *Umbilicosphaera* &*Calcidiscus*

[Figure]

*U.hulburtiana*

[Figure]

*U. foliosaU. sibogaeU.*sp. 1

[Figure]

*U.*sp. 1 cell collapsed

[Figure]

*U.*sp. 1 coccolith detached

[Figure]

*U.*sp. 2                          *C. leptoporus*

Plate Ⅳ.*Reticulofenestra&Ceratolithus&Pontosphaera&Discosphaera*

[Figure]

*Reticulofenestra* sp. 1    *Reticulofenestra* sp. 2

[Figure]

*C. cristatus* CER *telesmus* type*C. cristatus* HET coccolithomorpha type*P. syracusana*

[Figure]

*D. tubifera*

Plate Ⅴ. Syracosphaeraceae: *Syracosphaera*

[Figure]

Cell disintegrated*S. histrica*

Plate Ⅵ. Mixed group

[Figure]

Coccolith-missed coccosphere                    Cell collapsed

Unknownsp. 1          Unknown sp. 2          Unknown sp. 3

Unknownsp. 3                              Unknown sp. 4

Unknown sp. 4          Coccolith deformed

---

## Referee Comment (RC2) · Anonymous Referee #1 · 4 Jul 2017

[revised manuscript text omitted]

Plate Ⅱ. Umbellosphaeraceae: *Umbellosphaera*

[Figure]

*U. irregularis*

*U. irregularis*

*U. tenuis*          *U. tenuis* type I

Plate III. Calcidiscaceae: *Umbilicosphaera* & *Calcidiscus*

[Figure]

*U. hulburtiana*

*U. foliosa*

*U. sibogae*

*U.* sp. 1

*U.* sp. 1 cell collapsed

*U.* sp. 1 coccolith detached

*U.* sp. 2

*C. leptoporus*

Plate IV. *Reticulofenestra* & *Ceratolithus* & *Pontosphaera* & *Discosphaera*

[Figure]

*Reticulofenestra* sp. 1    *Reticulofenestra* sp. 2

*C. cristatus* CER *telesmus* type    *C. cristatus* HET coccolithomorpha type    *P. syracusana*

[Figure]

*D. tubifera*

Plate V. Syracosphaeraceae: *Syracosphaera*

[Figure]

Cell disintegrated      *S. histrica*

**Plate VI. Mixed group**

[Figure]

Coccolith-missed coccosphere

Cell collapsed

Unknown sp. 1

Unknown sp. 2

Unknown sp. 3

Unknown sp. 3

Unknown sp. 4

Unknown sp. 4

Coccolith deformed

---

## Author Comment (AC2) · 7 Jul 2017

[revised manuscript text omitted]

4). *F. profunda* was only distributed below 50 m at St. I316, indicating a stratified and stable water locally. In other words, this peculiar hydrology was not caused by vertical upwelling, maybe water mass advection instead. It is very hard to identify what kinds of currents created this peculiar biophysical distribution, after all water currents are not prosperous during intermonsoon period.

The POC pattern can be represented by coccosphere abundance. Varied allocation to calcification produced dissimilarities in the PIC/POC ratios. Large rain ratio values around the Sri Lanka waters predicted a mineral ballast with a drawdown of biological carbon towards the deep seafloor (Iglesias-Rodriguez et al., 2008; Findlay et al., 2011). We suggest that the rain ratio (Zondervan et al., 2002) is of great importance in predicting biominerolization and photosynthetic production (Bolton et al., 2016).

**4.2 Coccolithophore ecological preferences**

Many coccolithophore indicator species were collected in this study although several were uncommon. *G. oceanica* is a representative dominant species that shows preference for eutrophic water (Andruleit et al., 2000). In the surface distribution of *G. oceanica*, both coccoliths and coccospheres were predominantly distributed in the easternmost waters of Sri Lanka. This may be due to the nutrientseutrophic water derived from the highly productive Andaman Sea which was linked to the Bay of Bengal through narrow channels (Gibson et al., 2007; Nielsen et al., 2004). The coccosphere of *U. irregularis* was only common in the equatorial zone, indicating oligotrophic water conditions there (Kleijne et al., 1989). In the Indian Ocean, eight species of *Florisphaera* were discovered in deep water (Kahn and Aubry, 2012). We found only one species of *Florisphaera* (*F. profunda*) and it typically occurred in the disphotic layer below 100 m. As an inhabitant of deep water, *F. profunda* hardly occurred inwas not found in surface water layer unless with the appearance of vertical upwelling, indicating a stratified and stable water system. The coccoliths of *U. sibogae* and *H. carteri* maxima were found at St. I316 indicating that these species showed affinities to low temperature and high salinity water. 
[revised manuscript text omitted]

Baumann, K. H., Cepek, M., and Kinkel, H.: Coccolithophores as indicators of ocean water masses, surface-water temperature, and paleoproductivity—examples from the South Atlantic, in: Use of Proxies in Paleoceanography, 117-144, Springer, doi10.1007/978-3-642-58646-0_4, 1999.

Bolton, C. T., Hernández-Sánchez, M. T., Fuertes, M.-Á., González-Lemos, S., Abrevaya, L.,

Mendez-Vicente, A., Flores, J.-A., Probert, I., Giosan, L., and Johnson, J.: Decrease in coccolithophore calcification and $CO_2$ since the middle Miocene, Nat. Commun., 7, doi: 10.1038/ncomms10284, 2016.

Broerse, A., Brummer, G.-J., and Van Hinte, J.: Coccolithophore export production in response to monsoonal upwelling off Somalia (northwestern Indian Ocean), Deep-Sea Res. Pt. II: Topical Studies in Oceanography, 47, 2179-2205, 2000.

Brown, C., and Yoder, J.: Distribution pattern of coccolithophorid blooms in the western North Atlantic Ocean, Cont. Shelf Res., 14, 175-197, 1994.

Clarke, K. R., and Warwick, R.M.: Change in marine communities: an approach to statistical analysis and interpretation, Plymouth, UK: Primer    E, 2001.

Cox, M. A., and Cox, T.: Interpretation of Stress in non-metric multidimensional scaling, Statistica Applicata, 4, 611-618, 1992.

Cros, L., and Fortuño, J. M.: Atlas of northwestern Mediterranean coccolithophores, Sci. Mar., 66, 1-182, 2002.

Eppley, R. W., Reid, F., and Strickland, J.: Estimates of phytoplankton crop size, growth rate, and primary production, Calif. Univ. Scripps Inst. Oceanogr. Bull., 1970.

Feng, Y., Warner, M. E., Zhang, Y., Sun, J., Fu, F.-X., Rose, J. M., and Hutchins, D. A.: Interactive effects of increased pCO2, temperature and irradiance on the marine coccolithophore *Emiliania huxleyi* (Prymnesiophyceae), Eur. J. Phycol., 43, 87-98, doi: 10.1080/09670260701664674, 2008.

Ferreira, J., Mattioli, E., and Van de Schootbrugge, B.: Palaeoenvironmental vs. evolutionary control on size variation of coccoliths across the Lower-Middle Jurassic, Palaeogeogr. Palaeocl., 465, 177-192, doi: 10.1016/j.palaeo.2016.10.029, 2017.

Findlay, H. S., Calosi, P., and Crawfurd, K.: Determinants of the PIC: POC response in the coccolithophore *Emiliania huxleyi* under future ocean acidification scenarios, Limnol. and Oceanogr., 56, 1168-1178, doi:10.4319/lo.2011.56.3.1168, 2011.

Fink, C., Baumann, K.-H., Groeneveld, J., and Steinke, S.: Strontium/Calcium ratio, carbon and oxygen stable isotopes in coccolith carbonate from different grain-size fractions in South Atlantic surface sediments, Geobios, 43, 151-164, 10.1016/j.geobios.2009.11.001, 2010.

Gibson, R. N., Atkinson, R. J. A., & Gordon, J. D. M.: Coral reefs of the Andaman Sea—an integrated perspective. Oceanography and Marine Biology: An Annual Review, 45, 173-194, doi: 10.1201/9781420050943.ch5, 2007.

Giraudeau, J., and Bailey, G. W.: Spatial dynamics of coccolithophore communities during an upwelling event in the Southern Benguela system, Cont. Shelf Res., 15, 1825-1852, doi:10.1016/0278-4343(94)00095-5, 1995.

Guerreiro, C., Oliveira, A., De Stigter, H., Cachão, M., Sá, C., Borges, C., Cros, L., Santos, A., Fortuño, J.-M., and Rodrigues, A.: Late winter coccolithophore bloom off central Portugal in response to river discharge and upwelling, Cont. Shelf Res., 59, 65-83, doi: 10.1016/j.csr.2013.04.016, 2013.

[revised manuscript text omitted]

Smith, H. E. K., Poulton, A. J., Garley, R., Hopkins, J., Lubelczyk, L. C., Drapeau, D. T., Rauschenberg, S., Twining, B. S., Bates, N. R., and Balch, W. M.: The influence of environmental variability on the

biogeography of coccolithophores and diatoms in the Great Calcite Belt, Biogeosciences Discussions, 1-35, 10.5194/bg-2017-110, 2017.

Sun, J., and Liu, D.: Geometric models for calculating cell biovolume and surface area for phytoplankton, J. Plankton Res., 25, 1331-1346, doi: 10.1093/plankt/fbg096, 2003.

Sun, J.: Organic carbon pump and carbonate counter pump of living coccolithophorid, Advances in Earth Science, 22, 1231-1239, 2007.

Sun, J., Gu, X. Y., Feng, Y. Y., Jin, S. F., Jiang, W. S., Jin, H. Y., and Chen, J. F.: Summer and winter living coccolithophores in the Yellow Sea and the East China Sea, Biogeosciences, 11, 779-806, doi:10.5194/bg-11-779-2014, 2014.

Taylor, A. R., and Brownlee, C.: Calcification, in: The Physiology of Microalgae, 301-318, Springer International Publishing, 2016.

Taylor, A. R., Brownlee, C., and Wheeler, G.: Coccolithophore cell biology: chalking up progress, Ann. Rev. Mar. Sci., 9, 283-310, doi: 10.1146/annurev-marine-122414-034032, 2017.

[revised manuscript text omitted]

*E. huxleyi* type A overcalcified*G. oceanica*

[Figure]

*G. oceanica* coccolith

[Figure]

*G. oceanica* collapsed

Plate Ⅱ. Umbellosphaeraceae: *Umbellosphaera*

[Figure]

*U. irregularis*

[Figure]

*U. irregularis*

[Figure]

*U. tenuisU. tenuis* type Ⅰ

Plate Ⅲ. Calcidiscaceae: *Umbilicosphaera &Calcidiscus*

[Figure]

*U.hulburtiana*

[Figure]

*U. foliosaU. sibogaeU.*sp. 1

[Figure]

*U.*sp. 1 cell collapsed

[Figure]

*U.*sp. 1 coccolith detached

[Figure]

*U.*sp. 2                    *C. leptoporus*

Plate Ⅳ.*Reticulofenestra&Ceratolithus&Pontosphaera&Discosphaera*

[Figure]

*Reticulofenestra* sp. 1          *Reticulofenestra* sp. 2

[Figure]

*C. cristatus* CER *telesmus* type*C. cristatus* HET coccolithomorpha type*P. syracusana*

[Figure]

*D. tubifera*

Plate Ⅴ. Syracosphaeraceae: *Syracosphaera*

[Figure]

Cell disintegrated*S. histrica*

Plate Ⅵ. Mixed group

[Figure]

Coccolith-missed coccosphere

Cell collapsed

Unknownsp. 1

Unknown sp. 2

Unknown sp. 3

Unknownsp. 3

Unknown sp. 4

Unknown sp. 4

Coccolith deformed

---

## Referee Comment (RC3) · Anonymous Referee #2 · 1 Aug 2017

General

This paper attempts to describe floristic characteristics of coccolithophore assemblages in the eastern equatorial Indian Ocean during the spring intermonsoon period in relation with hydrographic conditions. Planktonic coccolithophores have not been well investigated in the Indian Ocean compared with those in the Atlantic and Pacific. However, this attempt is not successful, primarily due to methodological unclarity and failure to reach solid conclusion as described below.

My fundamental concern is on species identification and enumeration. Cell counting was made exclusively by light microscopy in the present study. It is well recognized

[Figure]

that some coccolithophore species can be identified by light microscopy, but others not. In particular, counting of small species requires use of SEM. For example, Gephyrocapsa oceanica was dominant (Table 1) and no other Gephyrocapsa species were mentioned in the paper. I wonder whether the other species were absent in the study area, overlooked or counted as G. oceanica. General practice of cell enumeration of coccolithophores is by the use of SEM, or careful comparison of images between SEM and light microscopy prior to cell counting using authors' own material. Even with the latter procedure, a certain number of small coccolithophore cells remain unidentified. A total of 26 species were identified in the present study (p. 4 l. 3). Then, a question can be raised how adequately unidentified cells by light microscopy were treated in data processing. Since procedures regarding this issue are not stated at all, I cannot judge the reliability of species identification from the information provided. The quality of floristic date is crucial in the present paper, and I cannot evaluate results of authors' statistical analysis without due explanation on this matter. Other issues on the methods are given below.

Another criticism is the lack of solid conclusions. While the paper aims to characterize the coccolithophore assemblage during the spring intermonsoon period, I do not see how similar and/or dissimilar the assemblage is between monsoon and intermonsoon periods, largely because comparison of authors' results with existing knowledge is superficial and in-depth analysis is lacking. Even though our knowledge of ecology of modern coccolithophores is limited in the Indian Ocean, key literature should be referred carefully at least, among which some essential papers are ignored such as Takahashi and Okada, 2000, Mari. Micropaleontol., 39, 73-86; Schiebel et al., 2004, Mar. Micropaleontol., 51, 345–371; Young et al., 2017, Proc. Int. Ocean Discov. Prog., 359-111. In addition, the text is not well organized, that is, the results and discussion sections are not well differentiated, and there are too many figures for the length of the both sections. This implies each topic is treated superficially and not adequately discussed. As a consequence, overall impression of the both sections is disjointed, and not well focused.

Specific

Abstract: The abstract is a simple list of authors' findings and should be much more focused.

Introduction: The introduction should state a rationale for the study: why the investigation during the spring intermonsoon period is needed. But, the current introduction is a simple and insufficient compilation of existing knowledge, and findings of several key papers as mentioned above are not considered.

Materials and methods

P2 L31: What does "initial' mean?

P2 L33: Seven depths were sampled. But, data from eight depths are plotted in Fig. 11.

P3 L 10 and L12: How was cellular dimension measured? Was measurement made on all cells, or selected individual cells of each species? If the latter, how many cells were measured?

P3 L12, L16: According to these lines, authors appeared to adopt published values to cell dimension of own material. However, this can cause potentially significant error (Smayda, 1978, Phytoplankton Manual, Unesco, p. 273-279). Slight errors in cellular dimension result in significant errors in volume calculation. I reserve to judge the suitability of the POC and PIC calculation.

P3 L15: Species name should be given.

Results: Too many graphs with insufficient explanation and interpretation have made the paper unfocused.

P3 L33: It is unclear whether advection or watermass extention from the data presented.

P4 L5: What is the purpose to show the plates? The plates are not needed for the current explanation. Methods on SEM are not given in the MM section.

P4 L8: I do not see what "frequency" means.

P4 L12: Authors should pay attention to significant digits.

P4 L12-18: Numbers are given for which depth?

P4 L15: F. profunda appears to be most predominant from Fig. 3.

P4 L35: Ecological significance of the ratio should be given. The ratio can be subject to sample handling, and I wonder how authors differentiate coccospheres and coccoliths from intermediate stages of coccolith aggregation between fragments of a collapsed coccosphere and a single isolated coccolith.

P5 L5: Section names should be used instead of use of "inner, outer"

Discussion: Detailed specific comments are not useful at this stage.